# A Unified Framework for Heterogeneous Semi-supervised Learning

## Abstract

In this work, we introduce a novel problem setup termed as Heterogeneous Semi-Supervised Learning (HSSL), which presents unique challenges by bridging the semi-supervised learning (SSL) task and the unsupervised domain adaptation (UDA) task, and expanding standard semi-supervised learning to cope with heterogeneous training data. At its core, HSSL aims to learn a prediction model using a combination of labeled and unlabeled training data drawn separately from heterogeneous domains that share a common set of semantic categories; this model is intended to differentiate the semantic categories of test instances sampled from both the labeled and unlabeled domains. In particular, the labeled and unlabeled domains have dissimilar label distributions and class feature distributions. This heterogeneity, coupled with the assorted sources of the test data, introduces significant challenges to standard SSL and UDA methods. Therefore, we propose a novel method, Unified Framework for Heterogeneous Semi-supervised Learning (Uni-HSSL), to address HSSL by directly learning a fine-grained classifier from the heterogeneous data, which adaptively handles the inter-domain heterogeneity while leveraging both the unlabeled data and the inter-domain semantic class relationships for cross-domain knowledge transfer and adaptation. We conduct comprehensive experiments and the experimental results validate the efficacy and superior performance of the proposed Uni-HSSL over state-of-the-art semi-supervised learning and unsupervised domain adaptation methods.

## 1 Introduction

Deep learning models, owing to their hierarchical learned representations and intricate architectures, have monumentally advanced the state-of-the-art across a myriad of tasks (LeCun et al., 2015). Nonetheless, the success of deep learning has been often contingent on the availability of copious amounts of labeled data. Data annotation, especially in specialized domains, is not only resource-intensive but can also entail exorbitant costs (Sun et al., 2017). Consequently, semi-supervised learning (SSL) has been popularly studied, aiming to successfully utilize the free available unlabeled data to help train deep models in an annotation efficient manner (Van Engelen & Hoos, 2020).

However, current SSL methods assume that the unlabeled and labeled data are sampled from similar (homogeneous) distributions (Oliver et al., 2018). Such an assumption presents substantial practical limitations to applying traditional SSL methods to a wide range of application domains, where labeled and unlabeled data can have different distributions. For example, in the field of medical imaging, it is common for labeled MRI scans to be sourced from state-of-the-art research hospitals, while an influx of unlabeled scans could emanate from a myriad of rural clinics, each with its distinct scanning equipment and calibration idiosyncrasies. Similar heterogeneity patterns manifest in domains like aerial imagery, wildlife monitoring, and retail product classification. In such settings, the challenge lies in leveraging the unlabeled data given its dissimilarity with its labeled counterpart.

Therefore, to address the current limitations of the traditional SSL, we propose a novel heterogeneous semi-supervised learning (HSSL) task, where the training data consist of labeled and unlabeled data sampled from different distribution domains. The two domains contain a common set of semantic classes, but have different label and class feature distributions. The goal of HSSL is to train a model using the heterogeneous training data so that it can perform well on a held-out test set sampled from both the labeled and unlabeled domains. Without posing distribution similarity assumptions

between the labeled and unlabeled data, HSSL is expected to be applicable to a broader range of real-world scenarios compared to standard SSL. This novel heterogeneous semi-supervised learning task however is much more challenging due to the following characteristics: (1) The domain gap, expressed as divergence between class feature distributions across the labeled and unlabeled domains, presents a significant impediment to model generalization and learning. (2) The absence of annotated samples from the unlabeled domain during training further compounds the complexity of the task. (3) Considering that the test set comprises samples from both domains, the devised solution methods need to accurately model the distributions inherent to each domain. It is imperative for the models to discern not only the domain from which a sample originates but also the specific semantic class it belongs to. This requires either an explicit or implicit methodology to categorize samples accurately with respect to both domain origin and semantic class categories, distinguishing the task from both conventional SSL and unsupervised domain adaptation (UDA)—traditional SSL overlooks the domain heterogeneity within both the training and testing data, whereas UDA exclusively concentrates on the unlabeled domain as the target domain (Ganin & Lempitsky, 2015; Long et al., 2015). Therefore, traditional SSL and UDA methods are not readily applicable or effective in addressing the proposed HSSL task. A recent work (Jia et al., 2023) has made an effort to expand the traditional SSL task beyond its homogeneous assumptions. However, the proposed solution method learns separately in different domains using distinct components where an off-the-shelf UDA technique is employed to generate pseudo-labels for the unlabeled samples, bypassing the opportunity to train a unified cohesive model that could harness insights from both domains. Additionally, their test set is restricted to a labeled domain, whereas in HSSL, the objective is to train a model that effectively generalizes across both labeled and unlabeled domains. This task presents a significantly more complex challenge, as it requires the model to adapt and perform accurately in a broader range of scenarios.

In this work, we propose a novel method, named as Unified framework for Heterogeneous Semi-Supervised Learning (Uni-HSSL), to address the HSSL problem. The proposed method learns a fine-grained classification model cohesively under a unified framework by amalgamating the labeled and unlabeled class categories within an extended and precisely doubled label space. The framework consists of three technical components designed to tackle the HSSL challenges: a weighted moving average pseudo-labeling component, a cross-domain prototype alignment component and a progressive inter-domain mixup component. The pseudo-labeling component leverages a weighted moving average strategy to assign and update pseudo-labels for the unlabeled data. In this manner, it generates smooth and adaptive assignment of pseudo-labels, reducing the potential pitfalls of oscillating updates or noisy label assignments which is crucial given the significant domain gap between labeled data and unlabeled data. The cross-domain prototype alignment ensures that the inherent semantic structures of similar classes across the labeled and unlabeled domains are aligned. This alignment of class-centric prototypes between domains leverages inter-domain semantic class relationships, enabling knowledge transfer from the labeled domain to the unlabeled domain. The progressive inter-domain mixup component generates new synthetic instances by interpolating between labeled and unlabeled samples bridging the gap between the two domains. By adopting a progressive augmentation schedule, it gradually adapts the model to the distribution of the unlabeled domain, facilitating a steady and reliable transfer of knowledge. Comprehensive experiments are conducted on several benchmark datasets. The empirical results demonstrate the efficacy and superior performance of our proposed unified framework against multiple state-of-the-art SSL and unsupervised domain adaptation baselines for HSSL.

## 2 RELATED WORKS

### 2.1 SEMI-SUPERVISED LEARNING

Existing SSL methods can be succinctly categorized into three predominant paradigms: regularization-centric techniques, teacher-student architectures, and pseudo-labeling methodologies. A significant trajectory in SSL research orbits around regularization-centric methods. These approaches elegantly modify the loss function by introducing supplementary terms, ensuring desired properties in the trained model. Specifically, the $\Pi$-model (Laine & Aila, 2017) and Temporal-Ensemble (Laine & Aila, 2017) are distinguished by their embrace of consistency regularization. Notably, Temporal-Ensemble harnesses an exponential moving average of the model's predictions. An alternative strand in SSL, teacher-student frameworks, train a student network to emulate the predictions of a teacher model using unlabeled instances. The seminal Mean Teacher (MT) (Tarvainen & Valpola, 2017)

stands out, employing an exponential moving average on its teacher counterpart. Concurrently, ICT (Verma et al., 2022) underscores consistent model predictions over interpolated unlabeled data points, aligning them with interpolated predictions. Pseudo-labeling emerges as a strategic approach to augment labeled datasets using the unlabeled data. Pioneering this direction, Pseudo-Label (Lee et al., 2013) generates model-based labels, selectively emphasizing high-confidence predictions. Methodologies like FixMatch (Sohn et al., 2020) set confidence boundaries, producing pseudo-labels for different augmentation levels. In a progressive shift, Dash (Xu et al., 2021) and FlexMatch (Zhang et al., 2021) adaptively modulate confidence boundaries. SimMatch (Zheng et al., 2022) utilizes similarity-based label propagation to improve pseudo-label quality. In a recent work by Jia et al. (2023), the challenge of limited labeled data in one domain alongside abundant unlabeled data in another was addressed, with testing in a set that shares same feature distribution as labeled domain. Rather than integrating insights from both domains into a unified model, their method leveraged off-the-shelf UDA techniques for pseudo-labeling and treated domains separately, sidestepping the intricacies as well as advantages of cross-domain modeling.

## 2.2 Unsupervised Domain Adaptation

Unsupervised domain adaptation aims at learning a target model given labeled data from a source domain and unlabeled data from a target domain. Typical deep UDA approaches can be categorized into three types, i.e. alignment-based, regularization-based and self-training-based methods. Alignment-based methods aim to reduce the cross-domain feature discrepancy with adversarial alignment (Ganin & Lempitsky, 2015; Long et al., 2018) and distance-based methods (Long et al., 2015; Shen et al., 2018; Chen et al., 2020; Phan et al., 2023). Regularization-based methods utilize regularization terms to thoroughly leverage knowledge from the unlabeled target data. Typical regularization terms include entropy minimization (Shu et al., 2018), virtual adversarial training (Shu et al., 2018), batch spectral penalization (Chen et al., 2019), batch nuclear-norm maximization (Cui et al., 2020), and mutual information maximization (Lao et al., 2021). Self-training-based methods explore effective pseudo-labeling strategies for unlabeled target data fitting, including confidence score threshold (Zou et al., 2019; Berthelot et al., 2021) and cycle self-training (Liu et al., 2021).

## 3 Method

### 3.1 Problem Setup

We consider the following Heterogeneous Semi-Supervised Learning (HSSL) setup. The training data consist of a set of labeled instances $\mathcal{D}_L = \{(\mathbf{x}_i^l, \mathbf{y}_i^l)\}_{i=1}^{N_l}$, where each instance $\mathbf{x}_i^l$ is annotated with a one-hot label indicator vector $\mathbf{y}_i^l$ with length $C$, and a set of unlabeled instances $\mathcal{D}_U = \{\mathbf{x}_i^u\}_{i=1}^{N_u}$. The labeled data and unlabeled data are from two different domains that have dissimilar label distributions such that $p_L(\mathbf{y}) \neq p_U(\mathbf{y})$ and heterogeneous class feature distributions such that $p_L(\mathbf{x}|\mathbf{y}) \neq p_U(\mathbf{x}|\mathbf{y})$, but share the same set of $C$ semantic classes. The goal is to train a prediction model using the available labeled set $\mathcal{D}_L$ and unlabeled set $\mathcal{D}_U$ so that it would generalize well on a held-out test set that is indistinguishably sampled from both the labeled and unlabeled domains.

### 3.2 Proposed Method

In this section, we present the proposed Uni-HSSL method, which tackles the $C$-class HSSL problem by combining the labeled and unlabeled class categories to a doubled label space and learning a fine-grained $2C$-class classification model under a unified framework, aiming to adaptively handle the heterogeneous distributions across domains and gain better generalization over test instances randomly sampled from both the labeled and unlabeled domains. The core idea centers on simultaneously facilitating effective knowledge transfer from the labeled domain to the unlabeled domain while also harnessing the potential and information within the unlabeled data.

We start by first pre-training a feature encoder and a $C$-class semantic classifier on the labeled dataset, which can be used to produce the initial pseudo-labels of the unlabeled training data and provide partial initialization for our Uni-HSSL model. Then the $2C$-class Uni-HSSL model, which consists of a feature encoder $f$ and a $2C$-class classifier $h$, will be learned within the proposed unified semi-supervised framework shown in Figure 1. The framework introduces three technical

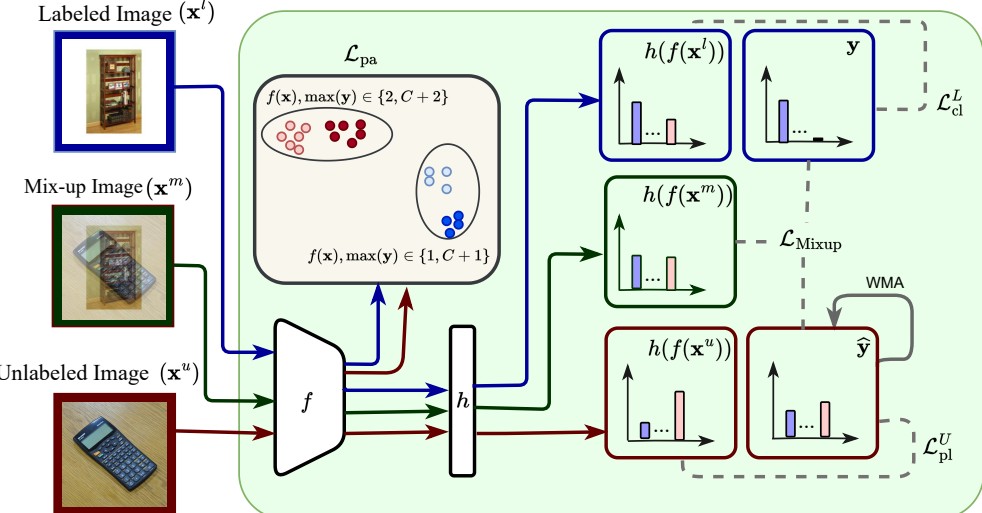

Figure 1: An overview of the proposed Uni-HSSL training framework. The classification model consists of a feature encoder $f$ and a $2C$-class classifier $h$. After initialization with pre-training, the model is trained by jointly minimizing the combination of a supervised loss $\mathcal{L}_{cl}^L$ on the labeled data, a WMA pseudo-labeling loss $\mathcal{L}_{pl}^U$ on the unlabeled data, a cross-domain prototype alignment loss $\mathcal{L}_{pa}$, and a prediction loss $\mathcal{L}_{\text{Mixup}}$ on the augmentation data produced via progressive inter-domain mixup.

components to facilitate heterogeneous SSL. The weighted-moving-average (WMA) based pseudo-labeling component is deployed to support the effective exploitation of the unlabeled data, while the cross-domain prototype alignment component and progressive inter-domain mixup component are designed to promote information sharing and efficient and steady knowledge transfer from the labeled domain to the unlabeled domain. Further elaboration will be provided in the following sections.

### 3.2.1 SUPERVISED PRE-TRAINING

The initial challenge in training a $2C$-class classification model with the given heterogeneous data is the absence of labeled instances entirely in the unlabeled domain. To tackle this problem, we exploit the assumption that the labeled and unlabeled domains share the same set of $C$ semantic class categories, and pre-train a $C$-class classification model in the labeled domain to provide initial pseudo-labels for the training instances in the unlabeled domain.

Specifically, we pre-train a $C$-class model, which consists of a feature encoder $f$ and a $C$-class probabilistic classifier $g$, on the labeled data $\mathcal{D}_L$ by minimizing the following supervised cross-entropy loss:

$$\mathcal{L}_{ce}^L = \mathbb{E}_{(\mathbf{x}_i^l, \mathbf{y}_i^l) \in \mathcal{D}_L}[\ell_{ce}(\mathbf{y}_i^l, g(f(\mathbf{x}_i^l)))] \tag{1}$$

where $\ell_{ce}$ denotes the cross-entropy function. Then we deploy the pre-trained classification model to make predictions on the unlabeled training instances in $\mathcal{D}_U$ to generate their initial pseudo-labels:

$$\bar{\mathbf{y}}_i^0 = g(f(\mathbf{x}_i^u)), \quad \forall \mathbf{x}_i^u \in \mathcal{D}_U \tag{2}$$

where $\bar{\mathbf{y}}_i^0$ denotes the predicted class probability vector with length $C$ for the unlabeled instance $\mathbf{x}_i^u$. To provide initial labels on the unlabeled data for training the $2C$-class model, we further expand each $\bar{\mathbf{y}}_i^0$ by concatenating it with a zero vector with length $C$, $\mathbf{0}_C$:

$$\hat{\mathbf{y}}_i^0 = \text{concat}(\mathbf{0}_C, \bar{\mathbf{y}}_i^0) \tag{3}$$

This results in the first set of $C$ classes out of the $2C$ classes corresponding to the classes in the labeled domain, with the remaining set of $C$ classes corresponding to the classes in the unlabeled domain. Moreover, the parameters of the pre-trained $C$-class model ($g \circ f$) can also be utilized to initialize the feature encoder $f$ and part of the classifier $h$ corresponding to the first $C$ classes in the $2C$-class model, while the other part of $h$ will be randomly initialized.

### 3.2.2 Semi-Supervised Training with Adaptive Pseudo-Labeling

After initialization, the proposed $2C$-class classification model (feature encoder $f$ and probabilistic classifier $h$) will be trained by leveraging both the labeled set $\mathcal{D}_L$ and the unlabeled set $\mathcal{D}_U$ within a pseudo-labeling based SSL framework. On the labeled set $\mathcal{D}_L$, the following standard supervised cross-entropy loss will be used as the minimization objective:

$$\mathcal{L}_{cl}^L = \mathbb{E}_{(\mathbf{x}_i^l, \mathbf{y}_i^l) \in \mathcal{D}_L}[\ell_{ce}(h(f(\mathbf{x}_i^l)), \text{concat}(\mathbf{y}_i^l, \mathbf{0}_C))] \tag{4}$$

where the concatenated label vector, $\text{concat}(\mathbf{y}_i^l, \mathbf{0}_C)$, expands the ground-truth label vector $\mathbf{y}_i^l$ into the $2C$-class label space by appending a zero vector with length $C$, $\mathbf{0}_C$, to it.

Although we have obtained initial pseudo-labels for the unlabeled set $\mathcal{D}_U$ by utilizing the pre-trained $C$-class classifier, those initial labels are unavoidably noisy due to the existence of domain gap between the labeled and unlabeled domains. In order to effectively leverage the unlabeled data, we update the pseudo-label for each unlabeled instance $\mathbf{x}_i^u$ during each training iteration in a weighted moving average (WMA) fashion as follows:

$$\hat{\mathbf{y}}_i^t = \beta \hat{\mathbf{y}}_i^{t-1} + (1 - \beta) h(f(\mathbf{x}_i^u)) \tag{5}$$

where $\beta \in (0, 1)$ is a hyper-parameter that controls the rate of update, and $\hat{\mathbf{y}}_i^t$ is the updated pseudo-label for $\mathbf{x}_i^u$ at the $t$-th training iteration. This weighted moving average update strategy can yield a smooth and adaptive assignment of pseudo-labels by promptly incorporating the progress in the classification model and mitigating the risk of oscillatory updates. Moreover, to further mitigate the adverse impact of noisy pseudo-labels, we deploy the following cross-entropy loss on the unlabeled set during training, selectively utilizing only instances with more reliable pseudo-labels:

$$\mathcal{L}_{pl}^U = \mathbb{E}_{\mathbf{x}_i^u \in \mathcal{D}_U}[\mathbb{1}(\max(\hat{\mathbf{y}}_i^t) > \epsilon) \ell_{ce}(h(f(\mathbf{x}_i^u)), \hat{\mathbf{y}}_i^t)] \tag{6}$$

where $\mathbb{1}(\cdot)$ denotes an indicator function; $\epsilon \in (0, 1)$ is a pre-defined confidence threshold to ensure that only unlabeled instances with the maximum prediction probabilities larger than $\epsilon$ are used for the current training iteration.

By treating semantic classes in distinct domains as separate categories, the $2C$-class classification model serves as a strategic choice to differentiate samples across domains. This approach avoids the additional complexity associated with a dedicated domain classifier and naturally handles the divergence in class-feature distributions across domains. It also simplifies the process and has the potential to enhance domain generalization through a shared feature encoder.

### 3.2.3 Cross-Domain Semantic Prototype Alignment

Given that the labeled domain and unlabeled domain are comprised of the same set of $C$ semantic classes, there is a one-to-one correspondence relationship between each cross-domain class pair for the same semantic concept. In order to facilitate knowledge sharing and transfer across domains, we propose to align each semantic class from the labeled domain with its corresponding semantic class in the unlabeled domain within the learned feature embedding space. To this end, we represent each class using a class-prototype vector and design a cross-domain semantic class-prototype alignment component to enforce that the corresponding semantic class pairs across the domains are more similar in the feature embedding space than non-corresponding class pairs.

Specifically, we compute the prototype vector for the $k$-th class in the labeled set as the average feature embedding of the labeled instances belonging to the class:

$$\mathbf{p}_k = \mathbb{E}_{(\mathbf{x}_i^l, \mathbf{y}_i^l) \in \mathcal{D}_L}\left[\mathbb{1}(\arg\max_j \mathbf{y}_{ij}^l = k) f(\mathbf{x}_i^l)\right] \tag{7}$$

where $\mathbf{y}_{ij}^l$ denotes the $j$-th entry of the label vector $\mathbf{y}_i^l$. The corresponding $k$-th semantic class in the unlabeled set is the $(C + k)$-th class in the $2C$-class label space. We compute the class prototype vectors in the unlabeled set based on the instances with reliable pseudo-labels, such that:

$$\mathbf{p}_{C+k} = \mathbb{E}_{\mathbf{x}_i^u \in \mathcal{D}_U}\left[\mathbb{1}(\max(\hat{\mathbf{y}}_i^t) > \epsilon \wedge \arg\max_j \hat{\mathbf{y}}_{ij}^t = C + k) f(\mathbf{x}_i^u)\right]. \tag{8}$$

Then for each semantic class $k \in \{1, \cdots, C\}$, we align the prototypes of the corresponding class pairs from the labeled and unlabeled domains, $(\mathbf{p}_k, \mathbf{p}_{C+k})$, by employing a cross-domain contrastive

prototype alignment loss as follows:

$$\mathcal{L}_{pa} = -\sum_{k=1}^{C} \left[ \log \frac{\exp(\cos(\mathbf{p}_k, \mathbf{p}_{C+k})/\tau)}{\sum_{k'=1}^{C} \mathbb{1}(k' \neq k)\exp(\cos(\mathbf{p}_k, \mathbf{p}_{C+k'})/\tau)} + \log \frac{\exp(\cos(\mathbf{p}_k, \mathbf{p}_{C+k})/\tau)}{\sum_{k'=1}^{C} \mathbb{1}(k' \neq k)\exp(\cos(\mathbf{p}_{k'}, \mathbf{p}_{C+k})/\tau)} \right] \tag{9}$$

where $\tau$ is a temperature hyper-parameter, and $\cos(\cdot, \cdot)$ denotes the cosine similarity function. This contrastive loss promotes the sharing of predictive information between the labeled and unlabeled domains by encouraging the corresponding class prototype pairs to be closer to each other while simultaneously pushing the non-corresponding cross-domain class prototype pairs farther apart.

### 3.2.4 PROGRESSIVE INTER-DOMAIN MIXUP

In order to bridge the gap between the labeled domain and the unlabeled domain, we propose a progressive inter-domain mixup mechanism to augment the training set by dynamically generating synthetic instances between the labeled set and unlabeled set, with the objective of facilitating steady and efficient knowledge transfer from the labeled domain to the unlabeled domain.

Specifically, we generate an inter-domain synthetic instance $(\mathbf{x}^m, \mathbf{y}^m)$ by mixing a labeled instance $(\mathbf{x}^l, \mathbf{y}^l)$ from the labeled set $\mathcal{D}_L$ with a pseudo-labeled instance $(\mathbf{x}^u, \hat{\mathbf{y}}^t)$ from the unlabeled set $\mathcal{D}_U$ through linear interpolation:

$$\mathbf{x}^m = \lambda \mathbf{x}^u + (1-\lambda)\mathbf{x}^l, \qquad \mathbf{y}^m = \lambda \hat{\mathbf{y}}^t + (1-\lambda)\operatorname{concat}(\mathbf{y}^l, \mathbf{0}_C), \tag{10}$$

where $\lambda \in [0,1]$ is the mixing coefficient. To fully utilize the available data in both domains, we can generate $N^m = \max(N^l, N^u)$ synthetic instances to form a synthetic set $\mathcal{D}_{\text{Mixup}}$ by mixing each instance in the larger domain with a randomly selected instance in the other domain.

In the standard mixup procedure (Zhang et al., 2018), the mixing coefficient $\lambda$ is sampled from a fixed $\text{Beta}(\alpha, \alpha)$ distribution with hyperparameter $\alpha$. To facilitate a steady and smooth adaptation from the labeled domain to the unlabeled domain for HSSL, we propose to dynamically generate the mixup data in each training iteration $t$ by deploying a progressive mixing up strategy that samples $\lambda$ from a shifted $\text{Beta}(\alpha, \alpha)$ distribution based on a schedule function $\psi(t)$, such that:

$$\lambda \sim \psi(t) \times \text{Beta}(\alpha, \alpha), \quad \psi(t) = 0.5 + \frac{t}{2T} \tag{11}$$

where $T$ denotes the total number of training iterations. Following this schedule, at the beginning of the training process, we have $\psi(0) \approx 0.5$ and $\lambda$ is sampled from the approximate interval $[0, 0.5)$ as the model prioritizes the labeled domain, guarding against noisy pseudo-label predictions from unlabeled data. As the training progresses, the model gradually increases its reliance on the unlabeled data, and the interval $[0, \psi(t)]$ from which $\lambda$ is sampled is expanded gradually towards $[0, 1]$ (with $\psi(T) = 1$), allowing it to adapt seamlessly between domains.

Following previous works on using mixup data (Berthelot et al., 2019), we employ the mixup set $\mathcal{D}_{\text{Mixup}}$ for model training by minimizing the following mean squared error:

$$\mathcal{L}_{\text{Mixup}} = \mathbb{E}_{(\mathbf{x}_i^m, \mathbf{y}_i^m) \in \mathcal{D}_{\text{Mixup}}} \left[ \|h(f(\mathbf{x}_i^m)) - \mathbf{y}_i^m\|^2 \right] \tag{12}$$

### 3.2.5 TRAINING OBJECTIVE

By integrating the classification loss terms on the labeled set, the unlabeled set, and the mixup set, with the class prototype alignment loss, we obtain the following joint training objective for the Uni-HSSL model:

$$\mathcal{L}_{total} = \mathcal{L}_{cl}^{L} + \lambda_{pl}\mathcal{L}_{pl}^{U} + \lambda_{pa}\mathcal{L}_{pa} + \lambda_{\text{Mixup}}\mathcal{L}_{\text{Mixup}} \tag{13}$$

where $\lambda_{pl}$, $\lambda_{pa}$ and $\lambda_{\text{Mixup}}$ are trade-off hyper-parameters. The training algorithm for the proposed method is provided in Appendix A.

## 4 EXPERIMENTS

### 4.1 EXPERIMENTAL SETUP

**Datasets** We conducted comprehensive experiments to evaluate the performance of our proposed framework on four image classification benchmark datasets: Office-31, Office-Home, VisDA, and

Table 1: Mean classification accuracy (standard deviation is within parentheses) on Office-31 dataset using ResNet-50 backbone. The first domain in each column indicates the labeled domain while the second domain indicates the unlabeled domain.

| | W/A | A/W | A/D | D/A | D/W | W/D | Avg. |
|---|---|---|---|---|---|---|---|
| Supervised | $68.6_{(1.6)}$ | $82.8_{(1.2)}$ | $85.1_{(1.8)}$ | $35.5_{(0.9)}$ | $96.9_{(0.4)}$ | $98.2_{(0.5)}$ | 77.8 |
| FlexMatch | $68.1_{(1.8)}$ | $81.3_{(1.3)}$ | $85.1_{(1.8)}$ | $63.0_{(2.1)}$ | $98.5_{(0.2)}$ | $98.9_{(0.2)}$ | 82.4 |
| FixMatch | $69.1_{(1.3)}$ | $83.4_{(0.9)}$ | $86.4_{(0.8)}$ | $53.7_{(1.3)}$ | $98.1_{(0.2)}$ | $98.2_{(0.2)}$ | 81.5 |
| SimMatch | $71.1_{(0.9)}$ | $84.1_{(1.0)}$ | $86.5_{(0.5)}$ | $68.6_{(1.1)}$ | $96.8_{(0.5)}$ | $98.8_{(0.4)}$ | 84.3 |
| CDAN+Sup | $61.2_{(1.2)}$ | $82.5_{(1.3)}$ | $87.4_{(2.2)}$ | $58.3_{(2.6)}$ | $79.2_{(0.4)}$ | $97.5_{(0.4)}$ | 77.7 |
| MCC+Sup | $71.5_{(2.7)}$ | $88.8_{(0.7)}$ | $89.1_{(0.5)}$ | $67.6_{(1.3)}$ | $81.7_{(0.7)}$ | $99.5_{(0.4)}$ | 83.0 |
| BiAdopt | $70.2_{(0.9)}$ | $85.0_{(0.5)}$ | $77.4_{(0.7)}$ | $67.1_{(1.0)}$ | $94.2_{(0.5)}$ | $98.5_{(0.3)}$ | 82.0 |
| Uni-HSSL | $\mathbf{73.1}_{(1.0)}$ | $\mathbf{90.2}_{(0.8)}$ | $\mathbf{90.0}_{(0.2)}$ | $\mathbf{72.1}_{(0.7)}$ | $\mathbf{100}_{(0.0)}$ | $\mathbf{100}_{(0.0)}$ | **87.5** |

Table 2: Mean classification accuracy (standard deviation is within parentheses) on Office-Home dataset using ResNet-50 backbone. The first domain in each column indicates the labeled domain while the second domain indicates the unlabeled domain.

| | A/C | C/A | C/R | R/C | R/A | A/R | A/P | P/A | C/P | P/C | P/R | R/P | Avg. |
|---|---|---|---|---|---|---|---|---|---|---|---|---|---|
| Supervised | $53.1_{(0.7)}$ | $66.0_{(1.2)}$ | $77.5_{(0.9)}$ | $63.9_{(1.2)}$ | $72.6_{(0.9)}$ | $75.1_{(0.7)}$ | $67.4_{(1.5)}$ | $69.1_{(1.0)}$ | $69.1_{(0.9)}$ | $64.6_{(0.9)}$ | $80.0_{(0.5)}$ | $77.9_{(0.1)}$ | 69.7 |
| FlexMatch | $51.1_{(1.2)}$ | $68.1_{(1.3)}$ | $72.1_{(0.9)}$ | $67.8_{(1.6)}$ | $59.0_{(1.2)}$ | $73.5_{(0.9)}$ | $64.0_{(0.9)}$ | $64.1_{(1.2)}$ | $65.6_{(1.0)}$ | $64.3_{(1.1)}$ | $73.3_{(0.7)}$ | $68.1_{(1.2)}$ | 65.9 |
| FixMatch | $51.9_{(1.5)}$ | $63.8_{(0.7)}$ | $79.5_{(0.5)}$ | $66.2_{(0.7)}$ | $74.1_{(0.5)}$ | $70.4_{(0.6)}$ | $62.7_{(0.6)}$ | $62.8_{(0.8)}$ | $65.1_{(1.1)}$ | $65.2_{(1.5)}$ | $78.1_{(0.4)}$ | $74.7_{(0.3)}$ | 67.9 |
| SimMatch | $57.8_{(1.6)}$ | $69.7_{(0.9)}$ | $78.5_{(0.5)}$ | $64.3_{(0.8)}$ | $70.5_{(0.5)}$ | $75.8_{(0.5)}$ | $68.9_{(0.6)}$ | $69.7_{(0.9)}$ | $70.0_{(0.4)}$ | $68.5_{(0.8)}$ | $78.1_{(0.2)}$ | $74.0_{(0.7)}$ | 70.5 |
| CDAN+Sup | $47.0_{(0.5)}$ | $63.9_{(0.7)}$ | $67.1_{(0.8)}$ | $67.0_{(1.2)}$ | $74.6_{(0.9)}$ | $66.5_{(0.8)}$ | $56.5_{(0.5)}$ | $74.9_{(1.2)}$ | $65.5_{(1.2)}$ | $66.8_{(0.6)}$ | $89.5_{(0.4)}$ | $78.2_{(1.3)}$ | 67.4 |
| MCC+Sup | $54.9_{(1.2)}$ | $70.5_{(0.3)}$ | $75.4_{(0.5)}$ | $69.3_{(0.5)}$ | $75.1_{(0.8)}$ | $77.3_{(0.8)}$ | $71.8_{(0.4)}$ | $76.1_{(0.2)}$ | $71.2_{(0.5)}$ | $68.0_{(0.5)}$ | $82.1_{(0.6)}$ | $77.0_{(0.2)}$ | 72.3 |
| BiAdopt | $55.1_{(1.8)}$ | $65.1_{(1.2)}$ | $75.2_{(1.2)}$ | $61.2_{(1.8)}$ | $69.1_{(1.9)}$ | $72.1_{(1.4)}$ | $64.9_{(1.3)}$ | $64.1_{(0.8)}$ | $69.1_{(1.4)}$ | $67.7_{(0.9)}$ | $76.2_{(1.2)}$ | $74.1_{(1.4)}$ | 67.8 |
| Uni-HSSL | $\mathbf{60.1}_{(0.9)}$ | $\mathbf{72.0}_{(0.7)}$ | $\mathbf{80.5}_{(0.4)}$ | $\mathbf{72.8}_{(0.6)}$ | $\mathbf{75.8}_{(0.6)}$ | $\mathbf{78.3}_{(0.5)}$ | $\mathbf{70.9}_{(0.8)}$ | $\mathbf{78.7}_{(0.4)}$ | $\mathbf{72.8}_{(0.7)}$ | $\mathbf{69.9}_{(0.9)}$ | $\mathbf{82.9}_{(0.4)}$ | $\mathbf{82.1}_{(0.5)}$ | **74.7** |

ISIC-2019. In all four datasets, we split the samples of each domain into 90/10 train/test data. **Office-31** Saenko et al. (2010) is comprised of a collection of 4,652 images spanning 31 different categories. The images are sourced from 3 distinct domains: Amazon (A), DSLR (D), and Webcam (W) with different image resolutions, quality, and lighting conditions. **Office-Home** (Venkateswara et al., 2017) is a large collection of over 15,500 images spanning 65 categories. The images are sourced from 4 diverse domains: Artistic images (A), Clip Art (C), Product images (P), and Real-World images (R). **VisDA-2017** (Peng et al., 2017) is a large-scale dataset tailored specifically for the visual domain adaptation task. This dataset includes images of 12 distinct categories from two domains, Synthetic (S) and Real (R). With the significant domain shift between the synthetic and real images, VisDA highlights the difficulties associated with bridging significant domain gaps. **ISIC-2019** is a comprehensive repository of skin cancer research images sourced from 4 different sources: BCN-20000 (BCN) (Combalia et al., 2019), Skin Cancer MNIST (HAM) (Tschandl et al., 2018), MSK4 (Codella et al., 2018), and an undefined source. We utilize BCN and HAM sources only as they include samples from all eight distinct classes, unlike the other two sources.

**Implementation Details** For all baselines we compared our Uni-HSSL against, we strictly followed the implementation details and hyper-parameters specified in the corresponding original papers. In order to ensure consistent comparisons with a multitude of earlier studies across various benchmark datasets, we employed two common backbone networks: ResNet-50 and ResNet-101 which are pre-trained on the ImageNet (Deng et al., 2009) dataset. We utilized ResNet-101 for VisDA dataset experiments and ResNet-50 for all the other benchmark datasets. The supervised pre-training stage is made up of 10 epochs while the semi-supervised training stage is made up of 100 epochs. In both stages, we employed an SGD optimizer with a learning rate of $5e^{-4}$ and Nesterov momentum (Nesterov, 1983) of 0.9. In the semi-supervised training stage, the learning rate is adjusted using a cosine annealing strategy (Loshchilov & Hutter, 2017; Verma et al., 2022). We set the L2 regularization coefficient to $1e^{-3}$ and the batch size to 32 for all datasets. The trade-off hyper-parameters $\lambda_{pl}$, $\lambda_{pa}$, $\lambda_{\mathrm{Mixup}}$ take the values 1, $1e^{-2}$ and 1 respectively, while $\tau$ and $\epsilon$ take the value 0.5 and $\beta$ is set to 0.8. Furthermore, we apply random translations and horizontal flips to the input images prior to applying the Progressive Inter-Domain Mixup similar to Berthelot et al. (2019). We report the mean classification accuracy and the corresponding standard deviation over 3 runs in each experiment.

Table 3: Mean classification accuracy (standard deviation is within parentheses) on VisDA dataset using ResNet-101 backbone. The columns correspond to the different classes of the dataset.

| | Plane | Bicyc. | Bus | Car | Horse | Knife | Motor. | Person | Plant | Skate. | Train | Truck | Avg. |
|---|---|---|---|---|---|---|---|---|---|---|---|---|---|
| Supervised | $93.8_{(0.2)}$ | $74.1_{(0.5)}$ | $79.4_{(0.7)}$ | $86.2_{(0.9)}$ | $90.9_{(0.2)}$ | $87.5_{(0.7)}$ | $94.5_{(0.4)}$ | $80.0_{(0.7)}$ | $91.1_{(0.7)}$ | $81.8_{(0.9)}$ | $96.0_{(0.3)}$ | $59.8_{(0.9)}$ | 84.1 |
| FlexMatch | $98.3_{(0.7)}$ | $74.8_{(0.9)}$ | $53.9_{(1.2)}$ | $36.4_{(2.1)}$ | $97.4_{(0.5)}$ | $77.2_{(0.8)}$ | $66.6_{(1.2)}$ | $80.5_{(0.8)}$ | $91.8_{(0.8)}$ | $90.0_{(0.5)}$ | $96.8_{(0.7)}$ | $49.2_{(1.2)}$ | 82.4 |
| FixMatch | $94.9_{(0.5)}$ | $53.5_{(0.2)}$ | $79.5_{(0.8)}$ | $88.5_{(0.3)}$ | $76.0_{(0.8)}$ | $78.8_{(0.9)}$ | $40.8_{(1.2)}$ | $58.9_{(1.6)}$ | $62.7_{(0.7)}$ | $68.9_{(1.2)}$ | $94.2_{(0.4)}$ | $49.5_{(1.2)}$ | 87.3 |
| SimMatch | $93.6_{(0.8)}$ | $81.1_{(0.8)}$ | $56.9_{(1.2)}$ | $59.6_{(1.5)}$ | $65.6_{(1.0)}$ | $71.9_{(0.5)}$ | $70.8_{(0.9)}$ | $64.1_{(0.8)}$ | $65.5_{(0.9)}$ | $57.0_{(1.7)}$ | $74.2_{(0.9)}$ | $52.1_{(1.7)}$ | 80.8 |
| CDAN+Sup | $98.4_{(0.3)}$ | $94.4_{(0.7)}$ | $90.1_{(0.5)}$ | $85.1_{(0.5)}$ | $96.6_{(0.1)}$ | $95.0_{(1.4)}$ | $96.6_{(0.5)}$ | $94.3_{(0.6)}$ | $96.5_{(0.4)}$ | $85.5_{(0.5)}$ | $95.7_{(0.7)}$ | $79.8_{(0.2)}$ | 92.1 |
| MCC+Sup | $\mathbf{98.6}_{(0.3)}$ | $96.6_{(0.5)}$ | $88.6_{(0.7)}$ | $84.8_{(0.9)}$ | $97.6_{(0.3)}$ | $95.1_{(0.9)}$ | $94.2_{(0.2)}$ | $94.6_{(0.2)}$ | $\mathbf{97.3}_{(0.5)}$ | $83.0_{(0.8)}$ | $95.6_{(0.1)}$ | $80.6_{(1.0)}$ | 92.0 |
| BiAdopt | $90.1_{(1.4)}$ | $79.1_{(1.2)}$ | $54.7_{(1.3)}$ | $56.1_{(1.2)}$ | $62.1_{(1.4)}$ | $68.2_{(0.1)}$ | $68.1_{(1.5)}$ | $62.5_{(1.2)}$ | $63.5_{(1.9)}$ | $59.3_{(1.7)}$ | $71.3_{(1.5)}$ | $50.1_{(1.5)}$ | 79.1 |
| Uni-HSSL | $98.2_{(0.5)}$ | $\mathbf{97.5}_{(0.9)}$ | $\mathbf{91.4}_{(0.8)}$ | $\mathbf{89.0}_{(0.9)}$ | $\mathbf{98.2}_{(0.3)}$ | $\mathbf{98.9}_{(0.4)}$ | $\mathbf{97.0}_{(0.6)}$ | $\mathbf{95.6}_{(0.7)}$ | $95.7_{(0.2)}$ | $\mathbf{91.5}_{(0.8)}$ | $\mathbf{97.0}_{(0.3)}$ | $\mathbf{82.4}_{(0.7)}$ | $\mathbf{93.1}$ |

## 4.2 COMPARISON RESULTS

We evaluate the proposed Uni-HSSL framework on the heterogeneous semi-supervised learning task and compare it to four baseline categories: Supervised, Semi-Supervised Learning (SSL), Unsupervised Domain Adaptation (UDA) baselines and Bidirectional Adaptation set up. The supervised baseline is exclusively trained on the labeled data and does not leverage the unlabeled data during training. We employ a set of representative SSL baselines (FlexMatch (Zhang et al., 2021), FixMatch (Sohn et al., 2020), and SimMatch (Zheng et al., 2022) ) and a set of representative UDA baselines (CDAN (Long et al., 2018) and MCC (Jin et al., 2020)). In addition we compare our work with BiAdopt Jia et al. (2023) As the traditional UDA methods are trained to perform well solely on an unlabeled target domain, to ensure a fair comparison, we equip the UDA methods with a Supervised classifier (Sup) trained on the labeled set and a domain classifier and refer to them as MCC+Sup and CDAN+Sup. The domain classifier assigns each test sample to the appropriate classifier at inference time; either the supervised classifier for samples predicted to originate from the labeled domain or the UDA classifier for those predicted to originate from the unlabeled domain.

The comparison results on Office-31, Office-Home, VisDA, and ISIC-2019 datasets are reported in Tables 1, 2, 3 and 4 respectively where the first domain indicates the labeled domain while the second domain indicates the unlabeled domain. In the case of VisDA dataset, the labeled dataset is sampled from the synthetic domain (S) and the unlabeled dataset is sampled from the real domain (R) and we report the average classification accuracy for each class and the overall average classification accuracy. The tables show that Uni-HSSL consistently outperforms all baselines on all datasets across all setups. The performance gains over the supervised baseline are notable exceeding 9%, 4%, 9%, and 7% on average in the cases of Office-31, Office-Home, VisDA, and ISIC-2019 datasets respectively. In the case of VisDA dataset, the performance improvement over the supervised baseline at the class level is substantial exceeding 22% for some classes.

Furthermore, Uni-HSSL consistently outperforms all SSL baselines, achieving performance gains exceeding 3%, 3%, 5%, and 3% over the most effective SSL baselines on Office-31, Office-Home, VisDA, and ISIC-2019 datasets, respectively. In some cases, such as A/W on Office-31 and P/A on Office-Home, the performance improvement over SSL baselines is notable surpassing 6% and 8% respectively highlighting the limitations of traditional SSL baselines in the proposed HSSL task. As for the case of UDA baselines, Uni-HSSL yields superior performance with all domain setups on all four datasets with performance gains around 4%, 2%, 1%, and 6% on Office-31, Office-Home, VisDA and ISIC-2019 datasets respectively. Uni-HSSL outperforms UDA baselines on almost all classes of VisDA dataset, with UDA baselines slightly excelling in only two classes. However, Uni-HSSL still maintains superior overall performance compared to both UDA baselines. Furthermore, MCC+Sup baseline does not perform well on ISIC-2019 dataset where it suffers a major drop in performance which can be attributed to MCC baseline's sensitivity to the class imbalance inherent in this dataset. Moreover, our Uni-HSSL outperforms BiAdopt with performance gains surpassing 5.5%, 6.9%, 14% and 4% on Office-31, Office-Home, VisDA and ISIC-2019 datasets, respectively. These results underscore the robustness of Uni-HSSL and highlight the limitations of BiAdopt in effectively addressing the challenges posed by the proposed HSSL task.

## 4.3 ABLATION STUDY

In order to investigate the contribution of each component of the proposed framework, we conducted an ablation study to compare the proposed Uni-HSSL with its six variants: (1) " −w/o WMA", which drops the Weighted Moving Average component of pseudo-label updates and simply uses

Table 4: Mean classification accuracy (standard deviation is within parentheses) on ISIC-2019 dataset using ResNet-50 backbone. The first domain in each row indicates the labeled domain while the second domain indicates the unlabeled domain.

| | Supervised | FlexMatch | FixMatch | SimMatch | CDAN+Sup | MCC+Sup | BiAdopt | Uni-HSSL |
|---|---|---|---|---|---|---|---|---|
| BCN/HAM | $70.5_{(0.9)}$ | $71.3_{(1.4)}$ | $77.5_{(0.8)}$ | $75.1_{(1.5)}$ | $72.9_{(1.0)}$ | $60.2_{(1.8)}$ | $74.2_{(0.7)}$ | $\mathbf{79.9}_{(0.7)}$ |
| HAM/BCN | $65.4_{(1.2)}$ | $68.7_{(0.8)}$ | $65.0_{(0.7)}$ | $69.2_{(1.7)}$ | $65.2_{(0.4)}$ | $56.7_{(1.7)}$ | $68.3_{(1.3)}$ | $\mathbf{71.0}_{(0.9)}$ |
| Avg. | 67.9 | 70.0 | 71.3 | 72.2 | 69.1 | 58.7 | 71.25 | **75.4** |

Table 5: Ablation study results in terms of mean classification accuracy (standard deviation is within parentheses) on Office-31 dataset using ResNet-50 backbone. The first domain in each column indicates the labeled domain while the second domain indicates the unlabeled domain.

| | W/A | A/W | A/D | D/A | D/W | W/D | Avg. |
|---|---|---|---|---|---|---|---|
| Uni-HSSL | $\mathbf{73.1}_{(1.0)}$ | $\mathbf{90.2}_{(0.8)}$ | $\mathbf{90.0}_{(0.2)}$ | $\mathbf{72.1}_{(0.7)}$ | $\mathbf{100}_{(0.0)}$ | $\mathbf{100}_{(0.0)}$ | **87.5** |
| $-$w/o WMA | $72.8_{(0.5)}$ | $87.1_{(0.8)}$ | $88.3_{(0.9)}$ | $71.0_{(0.8)}$ | $100_{(0.0)}$ | $100_{(0.0)}$ | 86.8 |
| $-$w/o $\mathcal{L}_{cl}^{L}$ | $67.6_{(1.7)}$ | $85.5_{(0.8)}$ | $86.1_{(1.2)}$ | $64.8_{(2.0)}$ | $93.2_{(0.5)}$ | $92.9_{(0.6)}$ | 81.7 |
| $-$w/o $\mathcal{L}_{pl}^{U}$ | $72.5_{(0.8)}$ | $87.9_{(0.9)}$ | $88.1_{(0.7)}$ | $71.0_{(0.9)}$ | $98.0_{(0.2)}$ | $98.5_{(0.2)}$ | 86.1 |
| $-$w/o $\mathcal{L}_{pa}$ | $72.7_{(0.5)}$ | $88.9_{(0.7)}$ | $87.2_{(0.6)}$ | $71.3_{(0.9)}$ | $99.1_{(0.0)}$ | $100_{(0.0)}$ | 86.5 |
| $-$w/o $\mathcal{L}_{\text{Mixup}}$ | $71.9_{(1.2)}$ | $86.7_{(0.9)}$ | $88.1_{(0.8)}$ | $71.3_{(1.1)}$ | $98.0_{(0.4)}$ | $99.9_{(0.0)}$ | 86.1 |
| $-$w/o Prog. Mixup | $71.3_{(0.9)}$ | $84.8_{(0.9)}$ | $88.1_{(1.0)}$ | $70.0_{(1.3)}$ | $99.2_{(0.5)}$ | $99.9_{(0.0)}$ | 85.6 |

the model predictions at the previous iteration to generate the pseudo-labels. (2) " $-$w/o $\mathcal{L}_{cl}^{L}$" drops the cross-entropy classification loss on the labeled set $\mathcal{D}_L$. (3) " $-$w/o $\mathcal{L}_{pl}^{U}$" drops the cross-entropy pseudo-label classification loss on the unlabeled set $\mathcal{D}_U$. (4) " $-$w/o $\mathcal{L}_{pa}$" drops the Cross-Domain Prototype Alignment component. (5) " $-$w/o $\mathcal{L}_{\text{Mixup}}$" drops the Progressive Inter-Domain Mixup component. (6) " $-$w/o Prog. Mixup" drops the progressive component of Inter-Domain Mixup and uses a simple mixup for inter-domain data augmentation. We compare the proposed Uni-HSSL with all its six variants on Office-31 dataset using ResNet-50 backbone and report the results in Table 5.

From the table, we can see that dropping any component from the proposed unified framework results in performance degradation in all cases. " $-$w/o $\mathcal{L}_{cl}^{L}$" variant suffered the largest performance degradation which highlights the importance of the ground-truth labels of $\mathcal{D}_L$ in guiding the learning process of the framework. Dropping the WMA from the pseudo-label generation component led to a slight average performance drop to 86.8%, underscoring its role in obtaining stable and confident pseudo-labels. Similarly, dropping the classification loss on the unlabeled data $\mathcal{L}_{pl}^{U}$ led to a perfor-mance degradation to 86.1%. Furthermore, the variant " $-$w/o Prog. Mixup" suffers a larger drop in performance in comparison with variant " $-$w/o $\mathcal{L}_{\text{Mixup}}$", this highlights the importance of progres-sively generating the augmented samples to ensure the accuracy of their corresponding augmented labels. Generating inter-domain augmented samples without taking into account the domain gap between the labeled domain and unlabeled domain can lead to a degradation in performance due to the noisy augmented labels of the generated samples. Overall, such consistent performance drops across all setups of Office-31 validate the essential contribution of each corresponding component of the proposed Uni-HSSL framework.

## 5 CONCLUSION

In this paper, we introduced a challenging heterogeneous semi-supervised learning problem, where the labeled and unlabeled training data come from different domains and possess different label and class feature distributions. To address this demanding setup, we proposed a Unified Framework for Heterogeneous Semi-Supervised Learning (Uni-HSSL), which trains a fine-grained classifica-tion model over the concatenated label space by effectively exploiting the labeled and unlabeled data as well as their relationships. Uni-HSSL adopts a WMA pseudo-labeling strategy to obtain stable and confident pseudo-labels for the unlabeled data, while deploying a cross-domain class prototype alignment component to support knowledge transfer and sharing between domains. A novel progressive inter-domain mixup component is further devised to augment the training data and bridge the significant gap between the labeled and unlabeled domains. We conducted comprehensive experiments on several benchmark datasets. The experimental results demonstrate the effectiveness and superiority of the proposed Uni-HSSL over state-of-the-art semi-supervised learning methods and unsupervised domain adaptation baselines.

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

---

**Algorithm 1** Training Algorithm for Uni-HSSL

---

**Input**: $\mathcal{D}_L, \mathcal{D}_U$; initialized prediction model $(f, h)$ and pseudo-labels $\{\hat{\mathbf{y}}_i^0\}_{i=1}^{N_u}$
**Output**: Trained feature encoder $f$ and $2C$-class classifier $h$
**for** t = 1 **to** T **do**
    Compute the supervised loss $\mathcal{L}_{cl}^L$ on $\mathcal{D}_L$ using Eq.(4)
    Update the pseudo-labels $\{\hat{\mathbf{y}}_i^t\}$ on $\mathcal{D}_U$ using Eq.(5)
    Compute the classification loss $\mathcal{L}_{pl}^U$ on $\mathcal{D}_U$ using Eq.(6)
    **for** each semantic class $k \in \{1, \cdots, C\}$ **do**
        Compute labeled class prototype $\mathbf{p}_k$ using Eq.(7)
        Compute unlabeled class prototype $\mathbf{p}_{C+k}$ using Eq.(8)
    **end for**
    Calculate contrastive prototype alignment loss $\mathcal{L}_{pa}$ using Eq.(9)
    Generate $\mathcal{D}_{\text{Mixup}}$ using Eq.(10), with $\lambda$ sampled via Eq.(11)
    Calculate the loss on the mix-up data $\mathcal{L}_{\text{Mixup}}$ using Eq.(12)
    $\mathcal{L}_{total} = \mathcal{L}_{cl}^L + \lambda_{pl}\mathcal{L}_{pl}^U + \lambda_{pa}\mathcal{L}_{pa} + \lambda_{\text{Mixup}}\mathcal{L}_{\text{Mixup}}$
    Update parameters of $f$ and $h$ using gradient descent.
**end for**

---

## A  TRAINING ALGORITHM FOR THE PROPOSED UNI-HSSL

The training algorithm for the proposed Uni-HSSL method is presented in Algorithm 1.

## B  ISIC DATASET

The details of BCN and HAM domains of ISIC-2019 dataset are presented in Table 6.

Table 6: The number of samples of each class in BCN and HAM domains of ISIC-2019 dataset.

|      | MEL   | NV    | BCC   | AK  | BKL   | DF  | VASC | SCC | Total  |
|------|-------|-------|-------|-----|-------|-----|------|-----|--------|
| BCN  | 2,857 | 4,206 | 2,809 | 737 | 1,138 | 124 | 111  | 431 | 12,413 |
| HAM  | 1,113 | 6,705 | 514   | 130 | 1,099 | 115 | 142  | 197 | 10,015 |

## C  HYPERPARAMETER SENSITIVITY

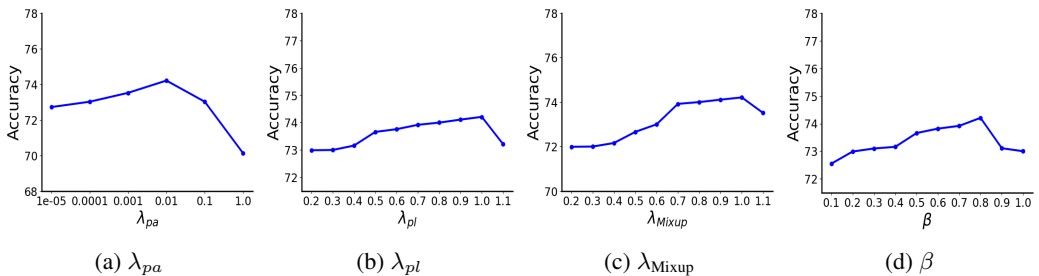

(a) $\lambda_{pa}$      (b) $\lambda_{pl}$      (c) $\lambda_{\text{Mixup}}$      (d) $\beta$

Figure 2: Sensitivity analysis for four hyper-parameters $\lambda_{pa}$, $\lambda_{pl}$, $\lambda_{\text{Mixup}}$, and $\beta$ on Office-31 using Webcam (W) as the labeled domain and Amazon (A) as the unlabeled domain.

We conduct sensitivity analysis for the proposed Uni-HSSL framework over four hyperparameters: the trade-off hyper-parameters controlling the contribution of each loss term $\lambda_{pa}$, $\lambda_{pl}$ and $\lambda_{\text{Mixup}}$ and $\beta$ hyperparameter controlling the rate of update for the pseudo-labels. We conducted the experiments on Office-31 using Webcam (W) as the labeled domain and Amazon (A) as the unlabeled domain by testing a range of different values for each of the four hyper-parameters independently. The obtained results are reported in Figure 2. From the figure, we can see that values either too small or too large cause performance degradation for the proposed framework for all hyper-parameters, while values in between yield the best results. In the case of the trade-off hyper-parameters controlling the

contribution of each loss term, this highlights the importance of the balanced interplay between the loss terms to obtain the best results without one loss term dominating the overall loss or one loss term having too small of a contribution. As for $\beta$, very large values prevent the framework from applying larger updates to the pseudo-labels in the early iteration of the training process, while very small values lead to oscillating updates across the training iterations. A value between 0.6 and 0.8 is required to obtain the best performance results.

