# OpenReview forum: "A Unified Framework for Heterogeneous Semi-supervised Learning"
_ICLR.cc/2024/Conference — Submitted to ICLR 2024_

### Official Review · Reviewer_Wmog · 2023-10-17

**Soundness:** 2 fair
**Presentation:** 2 fair
**Contribution:** 1 poor
**Rating:** 3
**Confidence:** 5

**Summary:**

This work introduces a novel problem setup termed Heterogeneous Semi-Supervised Learning (HSSL), where the labeled and unlabeled domains have dissimilar label distributions and class feature distributions.

**Strengths:**

The paper is generally written in a clear way.

**Weaknesses:**

The authors claim that they propose a novel method termed "Unified Framework for Heterogeneous Semi-supervised Learning (Uni-HSSL)". However, such setting has already been studied in previous works, such as "universal semi-supervised learning" (NeurIPS 21). The authors may not aware of this previous work, as they did not cite or compare this work. Therefore, I think the authors cannot claim that they "introduce a novel problem setup".

**Questions:**

I do not have specific questions.

---

> ### Author Response · Authors · 2023-11-20
> **Response to Reviewer Wmog**
>
> * **About the difference with Universal SSL**
>
> Our Heterogeneous Semi-supervised Learning (HSSL) setup introduces two key distinctions compared to the Universal Semi-supervised Learning setup:
>
> 1.  Our proposed setup aims to train a model capable of robust performance and generalization on a test set derived from **both the labeled and unlabeled domains**. In contrast, the test set in the Universal Semi-supervised Learning setup is **exclusively sampled from the labeled domain**, presenting a notably less challenging problem compared to our proposed setup.
>
> 2.  In HSSL, we posit that the labeled and unlabeled domains share an identical set of classes, yet exhibit dissimilar class label distributions. Furthermore, the class feature distributions for each semantic class are distinct. In contrast, Universal Semi-supervised Learning assumes a combination of shared/common and unshared/uncommon classes between the labeled and unlabeled domains.
>
> Consequently, we assert that our novel Heterogeneous Semi-supervised Learning (HSSL) task presents a more intricate and formidable challenge than the Universal Semi-supervised Learning setup.

---

### Official Review · Reviewer_uSh4 · 2023-10-26

**Soundness:** 2 fair
**Presentation:** 3 good
**Contribution:** 2 fair
**Rating:** 5
**Confidence:** 4

**Summary:**

This paper addresses a heterogeneous semi-supervised learning problem involving labeled and unlabeled data from different domains. The authors propose a framework called Uni-HSSL, which consists of three technical components: a weighted moving average pseudo-labeling component, a cross-domain prototype alignment component, and a progressive inter-domain mixup component. The proposed approach outperforms several SSL and UDA baselines on various benchmarks.

**Strengths:**

1. The problem being considered is interesting and important in real-world SSL applications.
2. The author has integrated several SSL technologies into a framework and in the experiments, the proposal has shown better performance compared to some baselines.
3. The overall proposal is well-presented and easy to follow.

**Weaknesses:**

1. The proposal seems to be a direct combination of existing technologies. The novelty of this proposal seems limited, and I am concerned that it may not bring new insights to the SSL community. The effectiveness of these techniques, such as using weighted moving averaging to reduce noise in pseudo-label updates, contrastive learning to strengthen prototype representation learning, and mixup to mitigate domain gaps, has been widely validated in the SSL/UDA community. It is foreseeable that combining them can improve the performance of SSL in a heterogeneous setting. However, the current version of the paper does not provide further analysis to explain or evaluate their effectiveness/reliability. More in-depth analysis, especially regarding their roles in heterogeneous SSL, can further improve this paper. For example, different technologies could be employed to handle noisy pseudo labels, such as ensemble, confidence-based selection, and entropy-based selection. Why did the current framework choose EMA, and what special capabilities does it have for heterogeneous SSL?
2. I also suggest that the author focus more on the issues in heterogeneous SSL rather than presenting the proposal from a technical perspective. From my understanding, in this article, the author uses three techniques to handle noisy pseudo labels and the misalignment of representation learning when facing cross-domain data. Defining the key problems in heterogeneous SSL may have a more positive impact on the community. For example, what difficulties do existing SSL techniques encounter due to the domain gap? Furthermore, the proposed Uni-HSSL achieves better heterogeneous SSL by addressing these problems separately.
3. In the experiment, the author only compared some previous baseline algorithms, and the SOTA method is missing. As mentioned by the author in the text, the ICML23 work considered the same problem, and it should be included in the comparison process of the experiment.

**Questions:**

1. The key problems in the heterogeneous SSL problem, and how to deal with these problems in the proposal? [See Weakness part]
2. How does the performance of Uni-HSSL compare to the ICML23 work?

Bidirectional adaptation for robust semi-supervised learning with inconsistent data distributions. ICML'23

---

> ### Author Response · Authors · 2023-11-20
> **Response to Reviewer uSh4**
>
> * **About novelty**
>
> Our work introduces a novel framework specifically tailored for HSSL, a task characterized by distinct challenges. While individual components, like weighted moving average pseudo-labeling, cross-domain prototype alignment, and inter-domain mixup, have been used in simpler setups, our innovation lies in their strategic integration and adaptation to the specific challenges of HSSL.
> Our approach introduces novel elements to prototype alignment by utilizing a $2C$ classifier, distinguishing it from conventional prototype alignment methods. Specifically, our innovation lies in the pseudo-labeling strategy where we consider the first $C$ classes in labeled data and the second $C$ classes in unlabeled data, seamlessly integrating with the $2C$ classifier framework. Additionally, our method redefines the mixup strategy by innovatively designing the updating strategy of the moving average decay parameter, which is designed to promote information sharing and efficient and steady knowledge transfer from the labeled domain to the unlabeled domain. As a result, our approach offers a novel perspective and a valuable contribution to addressing the intricacies of the proposed novel HSSL task.
>
> * **About presentation**
>
> We thank the reviewer for the suggestion about the paper organization and we will adjust the paper accordingly in the final version.
> This novel heterogeneous SSL task is much more challenging compared with the traditional SSL task due to the following characteristics: (1) The domain gap, expressed as the divergence between class feature distributions across the labeled and unlabeled domains, presents a significant impediment to model generalization and learning. (2) The absence of annotated samples from the unlabeled domain during training further compounds the complexity of the task. (3) Considering that the test set comprises samples from both domains, the devised solution methods need to accurately model the distributions inherent to each domain.
> Traditional SSL overlooks the domain heterogeneity within both the training and testing data, whereas UDA exclusively concentrates on the unlabeled domain as the target domain. Therefore, traditional SSL and UDA methods are not readily applicable or effective in addressing the proposed HSSL task
>
> * **About comparing to ICML23 work (BiAdopt)**
>
> Our Heterogeneous Semi-supervised Learning (HSSL) setup introduces a key distinction compared to the Semi-Supervised Learning with Inconsistent Data Distributions setup:
> Our proposed setup aims to train a model capable of robust performance and generalization on a test set derived from **both the labeled and unlabeled domains**. In contrast, the test set in the Semi-Supervised Learning with Inconsistent Data distribution setup is **exclusively sampled from the labeled domain**, presenting a notably less challenging problem compared to our proposed setup.
>
> Moreover, we have incorporated comparisons with BiAdopt in Tables 1, 2, 3, and 4. The updated tables showcase the superior performance of our proposed Uni-HSSL across all datasets. Notably, our Uni-HSSL outperforms BiAdopt with performance gains surpassing 5.5%, 6.9%, 14%, and 4% on Office-31, Office-Home, VisDA and ISIC-2019 datasets, respectively. These results underscore the robustness of Uni-HSSL and highlight the limitations of BiAdopt in effectively addressing the challenges posed by the proposed HSSL task.

---

> > ### Comment · Reviewer_uSh4 · 2023-12-01
> >
> > Thank you for the responses, which have addressed my concerns to some extent. After carefully considering the comments from other reviewers, I have decided to withhold my score.

---

### Official Review · Reviewer_xGwt · 2023-11-06

**Soundness:** 3 good
**Presentation:** 3 good
**Contribution:** 2 fair
**Rating:** 3
**Confidence:** 4

**Summary:**

This paper proposes a unified framework for Heterogeneous Semi-supervised Learning (Uni-HSSL), where the labeled and unlabeled data come from heterogeneous domains. It designs a weighted moving average pseudo-labeling component, a cross-domain prototype alignment component and an inter-domain mixup component to address the distribution inconsistency issue. The experiments validate the efficacy of the proposed framework.

**Strengths:**

1.	The paper is well-written and easy to follow.
2.	The paper solves semi-supervised learning under distribution inconsistency, an important ML problem in practice.
3.	Empirical results demonstrate that Uni-HSSL can achieve SOTA results on several benchmark SSL settings.

**Weaknesses:**

1.	The distribution mismatch between labeled and unlabeled data has been widely explored [1-5]. In this paper, it is crucial for the authors to discuss and compare these existing approaches to provide a comprehensive understanding of this field.
2.	The novelty is limited. This paper proposes three parts to address distribution mismatch issue: weighted moving average pseudo-labeling component, a cross-domain prototype alignment component and an inter-domain mixup component. However, the idea of moving pseudo-labels and mixup has been widely explored in semi-supervised learning [6][7]. And the prototype alignment is also widely used in UDA [8]. So in my opinion, this paper did not introduce new insight to SSL area.
3.	The paper only considers the DA dataset. I suggest authors could further investigate the effectiveness of their proposed framework in additional settings, such as imbalanced SSL with different imbalance ratios between labeled and unlabeled data on CIFAR10/100-LT benchmark.
4.	Some robust SSL methods are not compared, such as [5][9] in the experimental setting. And the authors only compare two UDA methods. The recent SOTA UDA methods [10] are missed.

[1] DC-SSL: Addressing Mismatched Class Distribution in Semi-Supervised Learning, CVPR 2022.

[2] DASO: Distribution-Aware Semantics-Oriented Pseudo-label for Imbalanced SSL, CVPR 2022.

[3] Class-Imbalanced Semi-Supervised Learning with Adaptive Thresholding, ICML 2022

[4] OpenMatch: Open-Set Semi-supervised Learning with Open-set Consistency Regularization, NeurIPS 2021.

[5] Universal Semi-Supervised Learning, NeurIPS 2021.

[6] Temporal Ensembling for Semi-Supervised Learning, ICLR 2017.

[7] MixMatch: A Holistic Approach to Semi-Supervised Learning, NeurIPS 2019.

[8] Weighted and Class-Specific Maximum Mean Discrepancy for Unsupervised Domain Adaptation, TMM 2020.

[9] Bidirectional adaptation for robust semi-supervised learning with inconsistent data distributions, ICML 2023.

[10] Patch-Mix Transformer for Unsupervised Domain Adaptation: A Game Perspective, CVPR 2023.

**Questions:**

See weakness for detail.

---

> ### Author Response · Authors · 2023-11-20
> **Response to Reviewer xGwt**
>
> * **About the novelty of the proposed HSSL**
>
> Heterogeneous Semi-Supervised Learning (HSSL) distinguishes itself from other SSL paradigms like Imbalanced SSL, Open Set SSL, and Universal SSL, each addressing distinct challenges in learning from limited labeled data.
>
> In **Imbalanced SSL**, the focus is on addressing disparities in class representation within a single domain; both labeled and unlabeled data **share the same domain**, implying equal conditional distributions ($P_l(\mathbf{x}|y) = P_u(\mathbf{x}|y)$). This contrasts with HSSL, which emphasizes learning under conditions of **domain differences**, where labeled and unlabeled data come from distinct domains with different label distributions and class feature distributions.
>
> **Open Set SSL** deals with **unknown or additional classes** present in the unlabeled data but absent in the labeled set. Open Set SSL also has the **same feature distribution** over labeled and unlabeled sets. This is different from HSSL, which operates under the assumption that both domains share **the same set of classes**, and labeled and unlabeled data come from separate domains with **different class feature distributions**.
>
> **Universal SSL** allows for a **mix of classes** in both labeled and unlabeled sets, some of which may not appear in the other. However, the **test set** maintains the same distribution as **the labeled set**. In contrast, HSSL maintains the assumption of **shared classes** across domains and more importantly, the **test set** includes instances sampled from **both labeled and unlabeled domains** which presents a more complex challenge.
>
> Each of these SSL paradigms addresses specific scenarios and challenges in semi-supervised learning, highlighting the diversity and complexity of learning with limited labeled data in various contexts.
>
>
> * **About novelty**
>
> Our work introduces a novel framework specifically tailored for HSSL, a task characterized by distinct challenges. While individual components, like weighted moving average pseudo-labeling, cross-domain prototype alignment, and inter-domain mixup, have been used in simpler setups, our innovation lies in their strategic integration and adaptation to the specific challenges of HSSL.
> Our approach introduces novel elements to prototype alignment by utilizing a $2C$ classifier, distinguishing it from conventional prototype alignment methods. Specifically, our innovation lies in the pseudo-labeling strategy where we consider the first $C$ classes in labeled data and the second $C$ classes in unlabeled data, seamlessly integrating with the $2C$ classifier framework. Additionally, our method redefines the mixup strategy by innovatively designing the updating strategy of the moving average decay parameter, which is designed to promote information sharing and efficient and steady knowledge transfer from the labeled domain to the unlabeled domain. As a result, our approach offers a novel perspective and a valuable contribution to addressing the intricacies of the proposed novel HSSL task.
>
> * **About using CIFAR10/100**
>
> Heterogeneous Semi-supervised Learning (HSSL) requires the labeled and unlabeled domains to have distinct feature distributions, necessitating the condition that $P_l(\mathbf{x}|y) \neq P_u(\mathbf{x}|y)$. However, conventional datasets like CIFAR typically represent a single domain and thus don't inherently meet this condition for HSSL, which requires distinct feature distributions across domains.
>
> * **About adding more comparisons**
>
> We have incorporated the comparison with [9] into our analysis. It is important to note that the PatchMix [10] utilizes the Transformer as its underlying backbone and it is not directly applicable to change its backbone therefore direct comparison is not suitable. Due to time constraints, we were unable to conduct experiments comparing our proposed method to [5] and [10] within the allotted rebuttal period. However, we commit to including these comparisons in the final version of the paper.

---

### Meta-Review · Area_Chair_otTs · 2023-12-14

**Metareview:**

This paper proposes a unified framework for Heterogeneous Semi-supervised Learning (Uni-HSSL), where the labeled and unlabeled data come from heterogeneous domains. It designs a weighted moving average pseudo-labeling component, a cross-domain prototype alignment component and an inter-domain mixup component to address the distribution inconsistency issue. The experiments validate the efficacy of the proposed framework.

Strengths:
1. The paper is well-written and easy to follow.
2. Empirical results show some effectiveness.
3. The proposed problem is interesting.

Weaknesses:
1. The paper borrows some technologies from existing SSL or UDA papers. Reviewers also raise their concerns about the novelty of the paper.
2. Some essential comparisons are missed.

**Justification For Why Not Higher Score:**

After reading the paper, I agree with most reviewers’ opinions about the novelty of the paper. The proposed modules show similarities to some previous papers. Further, the authors also acknowledge they still lack comparisons with some existing works. (e.g, reference paper [5] and [10] proposed by #Reviewer xGwt). I think these comparisons are essential to further show the effectiveness of the paper. I recommend the authors to further improve this work for a future submission.

**Justification For Why Not Lower Score:**

N/A

---

### Decision · Program_Chairs · 2024-01-16

Reject